# Airborne and Underwater Noise Produced by a Hovercraft in the North Caspian Region: Pressure and Particle Motion Measurements

Alexandr I. Vedenev [1,*], Oleg Yu. Kochetov [1], Andrey A. Lunkov [2] , Andrey S. Shurup [1,3,4]
and Saltanat S. Kassymbekova [5]

[1] Shirshov Institute of Oceanology, Russian Academy of Sciences, 36 Nakhimovsky Prospect, Moscow 117997, Russia; realspinner@gmail.com (O.Y.K.); shurup@physics.msu.ru (A.S.S.)
[2] Prokhorov General Physics Institute of the Russian Academy of Sciences, 38 Vavilov Str., Moscow 119991, Russia; lunkov@kapella.gpi.ru
[3] Schmidt Institute of Physics of the Earth of the Russian Academy of Sciences, 10 B. Gruzinskaya Str., Moscow 123242, Russia
[4] Faculty of Physics, Lomonosov Moscow State University, Leninskie Gory, Moscow 119991, Russia
[5] KMG Systems & Services LLP, 26-84 Koshkarbayev Str., Astana 010000, Kazakhstan; kassymbekova@kmgss.kz
[*] Correspondence: vedenev@ocean.ru

**Abstract:** The measurements of airborne and underwater noise radiated by a Griffon BHT130 hovercraft were conducted in the Ural-Caspian Channel and in the North Caspian Sea. This type of hovercraft is being used for all-season cargo and crew transportation to oil and gas platforms within the environmentally sensitive area of the Ural River estuary known for its abundant bird and fish fauna. Several field campaigns were organized from 2017 to 2022 to measure and analyze acoustic noise levels simultaneously in the air and underwater at various sites and hovercraft speeds. Airborne noise levels were estimated according to ISO 2922:2020, 2021. Underwater noise study included not only acoustic pressure recordings but also particle velocity measurements with a self-designed pressure gradient sensor (PGS), which is important since the hearing of the majority of fish perceives the sound in terms of particle motion. This study is the first to report the particle velocity levels formed underwater during hovercraft passages. The minimum levels of underwater noise, 100 dB re 1 µPa (pressure), 45 dB re 1 nm/s (particle velocity), and airborne noise, 93 dBA re 20 µPa (pressure), normalized to a distance of 25 m were observed for the hovercraft passages at a cruising speed of 7–15 m/s. Thus, this speed interval can be recommended as an optimum to minimize an acoustic impact on ornitho- and fish fauna. The directivity of the hovercraft noise was estimated for the first time and utilized for noise mapping of the Ural-Caspian Channel. The possible hydrodynamic effect of a passing hovercraft is discussed.

**Keywords:** hovercraft; shallow water; underwater noise; airborne noise; particle velocity; hydrodynamic pressure surge

## 1. Introduction

Soundscapes of anthropogenic underwater and airborne noise are studied extensively, especially within environmentally sensitive areas, to monitor the quality of fauna habitat and to study consequences of the man-made noise pollution. This particularly relates to the oil and gas fields [1–3]. The estuary of the Ural River and the Kazakhstan sector of the North Caspian are a unique wildlife preservation area and nesting area for rare birds, as well as a spawning place for valuable food fish can be attributed as one of such regions. In the meantime, the largest offshore oil field Kashagan is located in this region, where the oil companies have started using the hovercrafts on a regular basis for the transportation of personnel and cargo to a marine platform in the North Caspian Sea along the routes in the

estuary of the Ural River. A quite large hovercraft *Caspian Falcon*, that is shown in Figure 1, started running in October 2018. Specifications of this hovercraft are provided below.

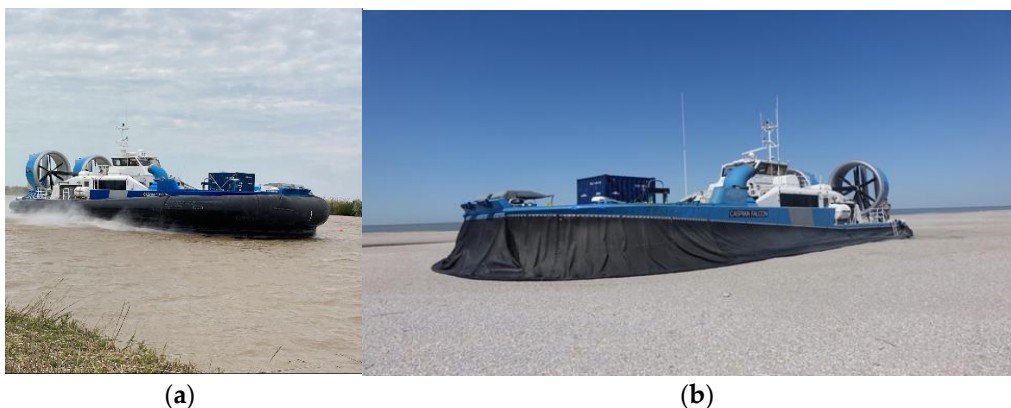

|   (**a**)   |   (**b**)   |

**Figure 1.** The hovercraft *Caspian Falcon* (**a**) in the Ural-Caspian Channel, (**b**) on a sand-spit in the Caspian Sea.

- Type: Griffon Hoverwork BHT130 hovercraft;
- Dimensions: Length: 30 m; beam: 13 m;
- Power system: Four diesel engines: two lift engines (900 HP each); two propulsion engines (1200 HP each);
- Propellers: Two 6-blade propellers.

In comparison with a small-sized Griffon 2000TD hovercraft, studied in [4], a Griffon BHT130 hovercraft is three times larger.

Negative underwater noise impact on wildlife depends on the type of noise (impulsive, transient, and continuous), its level and duration of exposure to noise. The majority of studies are carried out to assess the hazardous impact of impulse noise on aquatic wildlife during seismic surveying or pile driving [5]. Some of them included particle velocity measurements too [6,7]. For impulse noise, the thresholds of permissible maximum levels or duration of exposure have been established. Non-impulse (continuous) shipping noise can lead to behavioral effects on fish, such as elicitation of startle response, disruption of feeding, avoidance of an area, stress, masking of the communication signals, disturbance of the migration, and spawning. In case of critical levels or prolonged noise exposure, it can cause a temporary or permanent threshold shift of hearing (TTS and PTS) [8,9]. For the continuous noise impact on behavior, the exposure criteria have not been established yet by reason the noise levels upon which adverse impacts occur depend on many factors (fish species, age, sex, condition, and so on). Note that the fauna's behavioral reactions and noise exposure criteria are not within the scope of this paper where the study relates to hovercraft-emitted noise only.

The main goal of our work was to measure the parameters of the non-impulsive airborne and underwater noise from the hovercraft in order to evaluate the safety of using this type of vessel for ornitho- and fish fauna in a unique wildlife preservation area. The obtained acoustic data will be used for a further assessment of hovercraft noise impact on the wildlife in the North Caspian Sector.

The regular measurements of noise emitted by the hovercraft and other vessels were carried out in 2017, 2019, and 2022 [10–13] at various sites: the NCOSRB enclosed basin (the North Caspian Environmental Oil Spill Response Base), the Ural-Caspian Channel, and on the North Caspian shelf of Kazakhstan (see Figure 2). The measurement of hovercraft noise has the following features related to the area and period of field experiments:

A. A hovercraft's airborne noise has been measured in accordance with an ISO standard [14]. However, very shallow water (<5 m depth) conditions of the Ural-Caspian Channel and the North Caspian shelf do not fit the requirements of the ISO standard for underwater noise measurements. ISO 17208-1:2016 requires a water depth of

150 m [15]. The Ural-Caspian Channel has almost abrupt shores, an average depth of ≈5 m, width of ≈140 m, and river-specific muddy bottom. The water depth of the Caspian Sea near the Ural River estuary is ≈2 m. Therefore, the measured underwater noise parameters of the hovercraft relate to this specific region only. These very parameters of the hovercraft noise are required to assess the potential negative impact of the noise on the fish fauna within the environmentally sensitive shallow-water region. Exceeding the underwater noise precautionary threshold of 130 dB re 1 μPa has been considered as an onset of the negative behavioral response of fish [8,9,16,17].

B.  It was anticipated that underwater noise from a hovercraft would be lower and airborne noise would be higher than those of the vessels with marine screw propellers or water-jet propellers. The noise produced by the engine, propellers, and hull vibrations of such vessels is well transmitted to the water column and propagates there. Engines and propellers are the major sources of hovercraft noise [10–13,18]. However, once a hovercraft reaches cruising speed, the cushion contact with the water surface is almost negligible, which eliminates the direct transmission of noise and vibration from a hull into the water column. For a hovercraft, the only source of underwater noise is airborne noise, which is poorly transmitted through the air–water interface with reflection coefficient of 99.88% at normal incidence. Therefore, in contrast with ordinary vessels, a low level of underwater noise was expected during the hovercraft passage at a cruising and maximum speed (almost no contact of a skirt with the water's surface). A hovercraft in motion rises above the water's surface and generates noise which differs from the high-amplitude underwater noise of conventional ships with screw propellers. Its airborne noise is similar to that of aircrafts or helicopters as they all use air propellers that are the main source of airborne noise, which mainly reflects from air–water interface and weakly penetrates into water.

C.  Field experiments were carried out in the Ural-Caspian Channel (a man-made riverbed of the Ural River) in July-August 2017, August 2019, and May 2022 (Figure 2). The accuracy of noise measurements in the Spring of 2022 was limited due to the high wind speed (≥5 m/s) and strong current (up to 1 m/s) in the Ural River. In the meantime, nesting of rare bird species and fish spawning occurs during this very period. Low-frequency pseudo sound was produced by vortices around the PGS and affected the underwater noise recordings. Note that the main energy of pseudo sound is concentrated in the infrasound band. For the data collected at high current speeds, the infrasound part of noise spectra should be either filtered or simply ignored as born by non-acoustic phenomena. The latter was conducted. The frequency range from 100 Hz to 1650 Hz was chosen to calculate the particle velocity from the acoustic pressure time series.

D.  Particle velocity measurements of the hovercraft underwater noise were conducted using a self-designed pressure gradient sensor (PGS) which consisted of four non-coplanar hydrophones (Figure 3b). A finite-difference approximation of the spatial derivatives of the pressure field measured by these hydrophones makes it possible to assess three orthogonal projections of the particle velocity [18–21]. In [6], a similar PGS was used to study the impulse noise produced by pile driving. In [7], the particle acceleration sensors recorded underwater noise during seismic surveys. In [22], a PGS-type receiver was used to quantify ambient particle acceleration in coral reef soundscapes as well as for particle acceleration levels estimation due to the noise of boats.

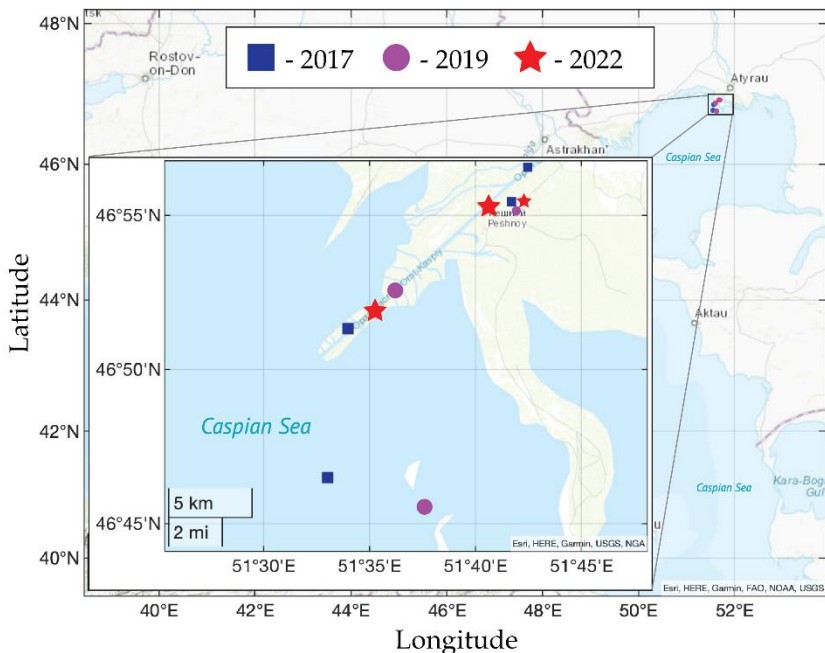

**Figure 2.** The North Caspian region with the sites of field measurements held in 2017, 2019, and 2022.

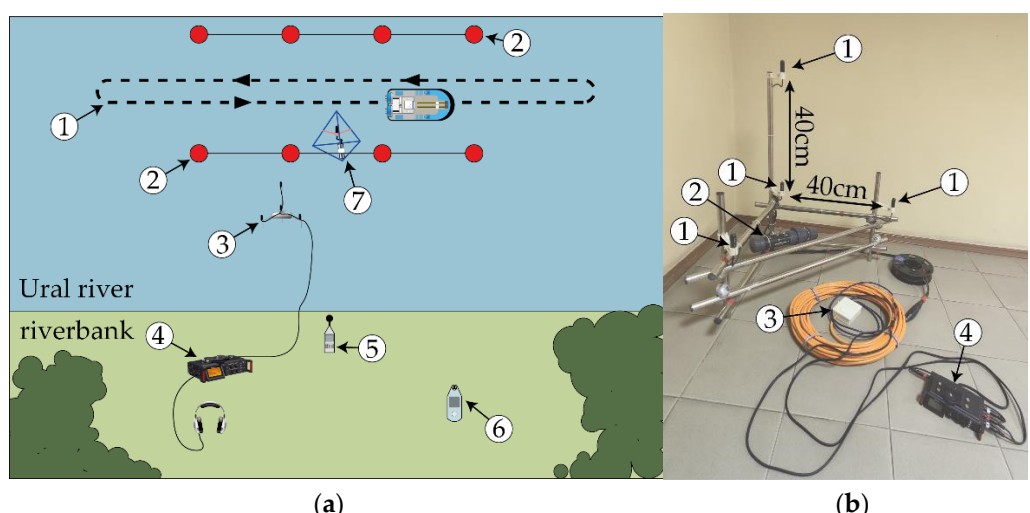

(**a**)                    (**b**)

**Figure 3.** (**a**) Experimental setup of the noise measurements in the Ural-Caspian Channel: (1) Hovercraft track; (2) Buoy lines; (3) PGS; (4) TASCAM DR-70D four-channel digital recorder; (5) NTM-Zashchita noise meter; (6) Anemometer; (7) RTsys autonomous underwater recorder with a hydrophone; (**b**) PGS with the signal recorder. (1) HTI-96min hydrophones; (2) Orientation logger; (3) Powered interface; (4) TASCAM DR-70D four-channel digital recorder.

This study is the first to report the particle velocity levels formed underwater during hovercraft passages. The importance of particle velocity measurements is determined by the fact that the hearing of the majority of fish species is sensitive to particle motion. The papers [8,9,16,17,23] provide a review of the latest scientific results on thresholds and acoustic impact of underwater sound on fish fauna in terms of particle velocity.

## 2. Materials and Methods

As an example of field procedures, a description of acoustic measurements in the Ural-Caspian Channel in May 2022 is presented. In that expedition, the widest range of acoustic equipment was used. Figure 3a shows the experimental setup of noise measurements. The hovercraft was running at different speeds between two buoy lines. Underwater noise was

recorded with the pressure gradient sensor (PGS) depicted in Figure 3b. RTsys autonomous underwater recorder equipped with a single hydrophone was deployed to measure the low-frequency noise. NTM-Zashchita noise meter with an MK-265 microphone and a windscreen was utilized to analyze the airborne noise.

The PGS consisted of four HTI-96min hydrophones and an orientation logger. It was connected to an onshore TASCAM DR-70D four-channel digital recorder with a 30 m waterproof cable. The sampling rate was 96 kHz. The steel frame of the PGS was additionally weighted with the lead plates to ensure the stable positioning on the bottom in conditions of a strong current. Hydrophones were placed in space along the orthogonal axes at a fixed distance of $h = 40$ cm. An orientation logger tracked the PGS frame position relative to the cardinal directions and the horizon. HTI-96min hydrophones had a sensitivity of -165 dB re 1 V/μPa.

An RTsys recorder with a calibrated hydrophone Bruel & Kjaer 8104 (sensitivity of $-205$ dB re 1 V/μPa) was integrated into a tetrahedron-shaped steel frame. The recorder was operating in a low frequency acquisition mode (<2 kHz).

To verify an equal sensitivity of PGS hydrophones, comparative measurements were carried out with a calibrated Bruel & Kjaer 8104 hydrophone. In these tests, 4 hydrophones HTI-96min forming PGS were tied up together with Bruel & Kjaer 8104 hydrophone. Then underwater test signals were recorded simultaneously by all hydrophones. Results of such comparative measurements showed that the sensitivity of PGS hydrophones corresponds to the manufacturer data in the frequency range of 10–2000 Hz. Moreover, the rms deviation of noise spectral density of PGS hydrophones from each other did not exceed 0.55 dB confirming their high identity, which indicates the possibility of their adequate use for particle velocity estimation.

The acoustical equipment for underwater measurements was deployed on the bottom in 5 m of water at a distance of 15–20 m from the shore. Soft fleecy protective covers were put on the hydrophones to minimize the pseudo sound evoked by the current and reduce acoustic bursts produced by collisions of suspended particles. A GPS tracker and a laser rangefinder were used to monitor the hovercraft position relative to the recording equipment. Wind speed was measured with an anemometer.

*2.1. Metrics of Underwater Noise*

There are a number of different acoustic metrics that could be used to assess the levels of underwater noise and its influence on the soundscape [5]. In our study, the following metrics are calculated from pressure time series $p_j(t)$ recorded at each hydrophone *j*:

- Time dependence of the sound pressure level (SPL)

$$SPL(t) = 20\lg\left(\frac{\sqrt{\frac{1}{\Delta t}\int_t^{t+\Delta t} < p^2(t') > dt'}}{p_0}\right), \tag{1}$$

measured in dB re $p_0 = 1$ μPa, $\Delta t = 1$ s is the averaging time, and $< .. >$ means an averaging over four hydrophones of PGS, i.e., $< p^2(t') > = \frac{1}{4}\sum_{j=1}^{4} p_j^2(t')$;

- Sound exposure level (SEL)

$$SEL = 10\lg\left(\frac{\int_T < p^2(t') > dt'}{p_0^2 T_0}\right), \tag{2}$$

measured in dB re $p_0^2 T_0 = 1$ μPa$^2$*s, and $T$ is the exposure time during which $SPL(t)$ is less than its maximum value by no more than 10 dB. In case of weak signals whose $SPL$ slightly exceeds the level of the ambient noise, the value of $T$ is chosen as the time interval at which $SPL$ is higher than the ambient noise level or levels of other acoustic events.

- Time dependence of the particle velocity level (VL)

$$VL(t) = 20\lg\left(\frac{\sqrt{\frac{1}{\Delta t}\int_t^{t+\Delta t} v^2(t')dt'}}{v_0}\right), \tag{3}$$

measured in the units of dB re $v_0 = 1$ nm/s. Here, $v(t') = \sqrt{v_x^2(t') + v_y^2(t') + v_z^2(t')}$ is the absolute value of particle velocity at time $t'$. Components $(v_x, v_y, v_z)$ of the particle velocity vector are calculated by taking the difference of the sound pressure at the relevant pairs of hydrophones of PGS. For example, the projection of particle velocity $v_{12}(t)$ onto the line 1–2, connecting two hydrophones (#1 and #2) is defined by the formula:

$$v_{12}(t) = -\frac{1}{\rho h}\int_0^t \left(p_1(t') - p_2(t')\right)dt', \tag{4}$$

where $\rho$ is the density of water; $h$ is the spacing between hydrophones #1 and #2. In the frequency domain, Equation (4) takes the form [20]:

$$v_{12}(f) = \frac{i}{2\pi f \rho}\frac{p_1(f) - p_2(f)}{h}, \tag{5}$$

where $i$ is the imaginary unit.

In order to ensure an adequate estimate of the particle velocity by (4) or (5), it is required to choose the proper frequency range. The upper-frequency limit is determined by the spacing $h$ between PGS hydrophones. Roughly, $h$ should not exceed a half wavelength (for more accurate estimation see [6,19]. The spacing $h$ is equal to 40 cm, which leads to the upper limit of $f \approx 1650$ Hz. In this case, amplitude error of finite-difference approximation (4) or (5) does not exceed 3 dB [6,19]. The lower frequency limit of the PGS operating range is dependent on (1) flow-induced noise at a low frequency due to a strong current in the Ural-Caspian channel; (2) the waveguide cutoff frequency; (3) a non-physical increase of noise level due to the division by small values of frequency in (5) (in this sense, the direct measurement of particle acceleration with a vector sensor has an advantage since this sensor has low self-noise at low frequency [24]). Eventually, the frequency range from 100 Hz to 1650 Hz is chosen to calculate the particle velocity from the acoustic pressure time series. Examples of particle velocity estimates are presented in Figures 4 and 5 and discussed in the next section.

The value of SEL and maximum levels of SPL and VL are normalized to a reference distance of 25 m in order to compare the measured levels of the hovercraft noise with the literature data, other vessels or between different passages. To calculate the normalized sound pressure level $SPL_{25}$, the spherical decay law is used:

$$SPL_{25} = SPL(t_{cpa}) + 20\lg\left(\frac{r_{cpa}}{25}\right), \tag{6}$$

where $t_{cpa}$ and $r_{cpa}$ are the values of time and range at the closest point of approach (CPA). The horizontal range $r_{cpa}$ is found between the hydrophone and the side of a vessel. Note that the spherical spreading of an acoustic wave recorded by a bottom-mounted hydrophone was verified in field experiments with a wideband transducer (See Supplementary Material). Normalized to 25 m values of $SEL$ and $VL$, i.e., $SEL_{25}$ and $VL_{25}$, are obtained assuming the same decay law.

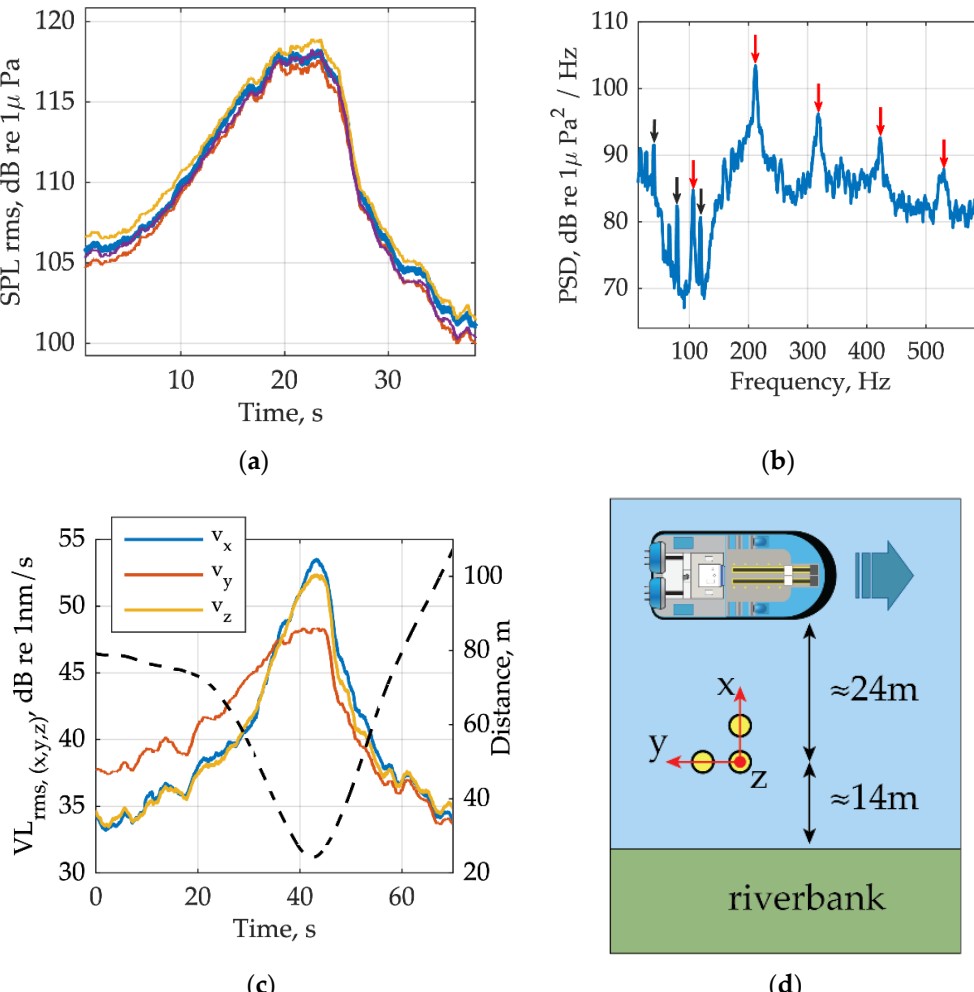

**Figure 4.** (**a**) Time dependence of *SPL(t)* in dB re 1 μPa at four hydrophones of PGS during one hovercraft passage (different colors correspond to different hydrophones). (**b**) The power spectral density of noise recorded by hydrophone located furthest from the bottom. The propeller blade frequencies are marked by red arrows; presumably shafting frequencies are marked by black arrows. (**c**) Time dependent $VL_{x,y,z}$ in dB re 1 nm/s calculated separately for three orthogonal components of particle velocity $(v_x, v_y, v_z)$ (different colors correspond to different components); black dashed line denotes the time dependence of the distance between PGS and the hovercraft. (**d**) Distances and orientation of PGS during measurements: red arrows and points show the direction of PGS axes, the hovercraft heading is marked by blue arrow.

## 2.2. Airborne Noise Measurement Methods

In contrast to underwater noise, the in-air internal and external noise of hovercrafts have been being measured and analyzed since the beginning of their operation in 1960s [4,25–27]. Note that, traditionally, the airborne sound of vessels is studied to evaluate the noise impact on humans. In recent years, investigations of airborne noise have been mainly dedicated to airborne sound emitted by moving ships and port activities recognized as disturbing sources on citizens [28–30]. If the underwater hovercraft noise level is less than that of other types of vessels due to the absence of submerged parts, the airborne noise is significantly higher and similar to that produced by a small propeller aircraft. Note that the discrete components corresponding to the propeller blade rate can be clearly distinguished in the noise of a hovercraft [4].

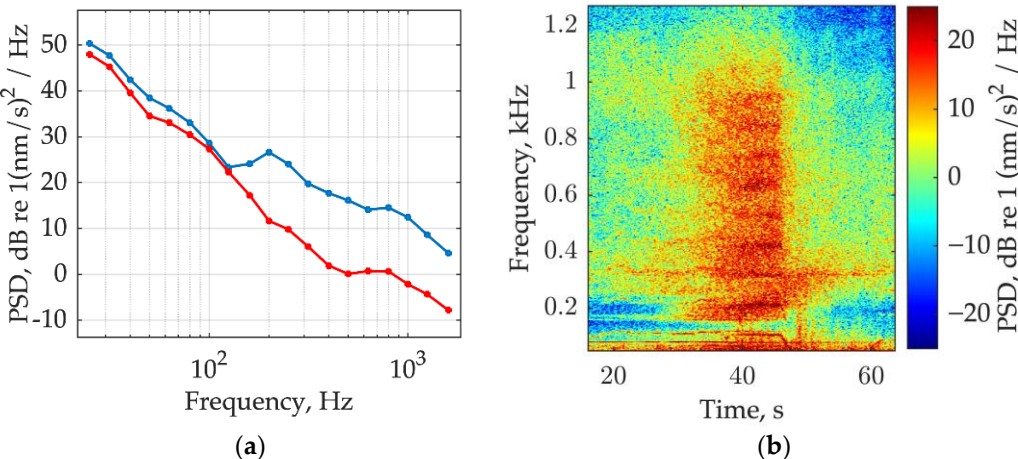

**Figure 5.** (**a**) Averaged power spectral density of particle velocity in the third-octave bands in the presence (blue line) and in the absence (red line) of the hovercraft. (**b**) Spectrogram of the vertical component $v_z$ of particle velocity.

Airborne noise from the hovercraft was measured using a NTM-Zashchita noise meter with a MK-265 microphone, which were mounted on a tripod on the shore (measurements on the river) or on an auxiliary vessel (measurements in the sea), 2 to 4 m above the water surface. The CPA was 30 to 60 m ($r_{cpa}$) from the microphone. The measurements of two metrics, sound pressure level (SPL), and sound exposure level (SEL), followed the recommendations of the ISO standard [14]. Time series of the 1 s-averaged A-weighted broadband (20–20,000 Hz) level, $SPL_A(t)$, and were recorded at every passage of the hovercraft. Sound exposure levels, $SEL_A$, are calculated over a period, when $SPL_A$ differs from its maximum value by no more than 10 dBA.

To compare the recorded noise levels between different passages and with levels of other vessels, the level of $SPL_A$ is normalized to a distance of 25 m using the expression similar to Equation (6):

$$SPL_{A25} = SPL_A(t_{cpa}) + 20\lg\left(\frac{r_{cpa}}{25}\right).$$ (7)

In (7), the spherical spreading of acoustic waves is assumed; $t_{cpa}$ and $r_{cpa}$ are the values of time and range at the point of closest approach, respectively. Sound exposure levels are normalized using the same approach. Results of airborne noise measurements are presented in Section 3.6.

## 3. Results

### 3.1. Measured Underwater Sound Pressure and Sound Exposure Levels of the Hovercraft

Figure 4a shows an example of $SPL(t)$ at four hydrophones of PGS during one passage of the hovercraft. The measured levels at four hydrophones are close to each other (the RMS deviation of SPL does not exceed 0.7 dB) that confirms the consistency of all four PGS channels. A similar situation is observed in other passages.

For the considered example, the maximum value of $SPL$ at individual hydrophones at CPA is 119 dB re 1 µPa. The value of $SEL$ averaged over four hydrophones is 128.2 dB re. 1 µPa$^2$ s. Levels of $SPL$ and $SEL$ are presented for the frequency band of 10 Hz–22 kHz.

Figure 4b shows the power spectral density of noise in units of dB re 1 µPa$^2$/Hz, recorded by one hydrophone located at the maximum distance from the bottom. A sharp decrease in the noise level at a frequency below 200 Hz is associated with the waveguide effect. According to the results of the acoustic experiments with a transducer and analysis of the bottom soil samples (see the Supplemental Material [31–33]), the bottom at experimental sites is acoustically soft, with a very low sound speed. In this case, an estimate of waveguide

cutoff frequency is $f_c = c/2H$, where $H$ is a depth of the water layer. In the conditions of the experiments held in the Ural River, the depth is $H \approx 4.5$ m, which gives an estimate for the cutoff frequency $f_c \approx 160$ Hz.

Noise produced by the thrust propellers is the major source of the hovercraft underwater and airborne noise. The airborne noise is generated directly by these propellers. Underwater noise occurs due to penetration of the airborne noise into the aquatic medium. The critical angle at the air–water interface is $\arcsin \frac{c_a}{c_w} \approx 13^\circ$, where $c_a \approx 343$ m/s is the sound speed in the air, and $c_w \approx 1500$ m/s is the sound speed in water. For the CPA in Figure 4c, the angle of noise arrival in a vertical plane in water can be estimated from particle velocity vector projections $(v_x, v_y, v_z)$ as $\arctan \frac{v_z}{\sqrt{v_x^2 + v_y^2}} \approx 53^\circ$, which corresponds to the angle of incidence of $\approx 11^\circ$ at the air–water boundary. This angle is less than the critical angle of $\approx 13^\circ$. Before and after passing the CPA, the hovercraft underwater noise undergoes multiple reflections from the bottom and water surface, which leads to a rapid decrease of the recorded noise level (see Figure 4a).

The quasi-periodic components corresponding to the propeller blade rate of the hovercraft are clearly seen in Figure 4b (marked with red arrows). Such a strong manifestation of the airborne noise underwater has been observed earlier in [4,10,20]. In Figure 4b, the fundamental frequency is $f_0 \approx \alpha f_d N/60 \approx 107$ Hz, where $f_d \approx 1600$ rounds per minute is a rotational frequency of the propulsion engine, $N = 6$ is a number of blades, and $\alpha = 1/1.5$ is a transmission coefficient from propulsion engine to blades. The value of $f_0$ is below the waveguide cutoff frequency, though this frequency is clearly distinguished in Figure 4b. Moreover, its harmonics $f_n \approx 213$ Hz, 319 Hz, 427 Hz are also present. Fundamental frequency of the blade shaft, $f_p \approx \alpha f_d/60 \approx 20$ Hz, can be noted in the low-frequency range, as well as its harmonics $\approx 40$ Hz, 80 Hz and 120 Hz (black arrows in Figure 4b). Low frequencies are also characterized by the other resonance frequencies, which apparently correspond to the rotation of the hovercraft lift engines whose detailed parameters remain unknown.

*3.2. Particle Velocity Estimates of the Hovercraft Underwater Noise*

Figure 4c shows particle velocity levels $VL_{x,y,z}(t)$ calculated separately for three orthogonal components $(v_x, v_y, v_z)$ from pressure measurements at four hydrophones of PGS, whose levels are illustrated in Figure 4a. The dashed line in Figure 4c denotes the time dependence of distance between PGS and the hovercraft. The orientation of PGS axes and distance to the CPA are demonstrated in Figure 4d. When the hovercraft is approaching PGS, $y$-projection of particle velocity, and $v_y$, has the highest level. At CPA, the maximum level is observed for $v_x$ component. In both cases, the highest level is obtained for the velocity projection closest to the direction towards the hovercraft position. Some unexpected result is the non-symmetry of $VL_{x,y,z}(t)$ before and after passing the CPA (see Figure 4a,c). It can be associated with the directivity of the hovercraft noise in a horizontal plane, which is not monopole-type. This effect will be discussed in Section 3.3.

Note that at the CPA, the level $VL_z$ (of vertical component $v_z$) is less than the level $VL_x$ (of horizontal component $v_x$) (Figure 4c). This result is expected since the vertical component of particle velocity is strongly dependent on bottom parameters in shallow water and in many cases attenuates faster than horizontal components [34,35]. However, an opposite situation may be observed, for example, during pile driving, where the vertical component of $v_z$ dominates [36]. Excitation of surface-type waves during the pile driving is the possible reason for this phenomenon [36,37].

The averaged power spectral density (PSD) of particle velocity Is shown in Figure 5a. Averaging is performed over all three components $(v_x, v_y, v_z)$. Blue line and red line denote the noise level in the presence and in the absence of the hovercraft, respectively. In contrast to Figure 4b, PSD calculations are made in third-octave bands. In Figure 5b, the spectrogram of vertical component $v_z$ is shown, which reveals sub-harmonics of propulsion engine frequencies as in Figure 4b. Moreover, a dip at low frequency can also be seen below 200 Hz.

In some cases, absolute value of particle velocity can be easily calculated as pressure $p$ divided by acoustic impedance $Z = \rho c$, where $\rho$ and $c$ are the known values of density and sound speed in water. It is known [16,20,23] that the plane wave approximation and a monopole sound source are necessary to make such recalculation possible. An estimate of the particle velocity level $\hat{V}L$ can be obtained from the pressure level $SPL$ according to the following formula

$$\hat{V}L = SPL - 20\lg(\rho c) + 60, \tag{8}$$

where the last term adjusts the normalization value when converting dB re 1 µPa into dB re 1 nm/s. An example of using Equation (8) is shown in Figure 6. One can see that the deviations between $\hat{V}L$ and $VL$ do not exceed $\approx 3$ dB, and near the CPA, the match between $\hat{V}L$ and $VL$ is almost exact.

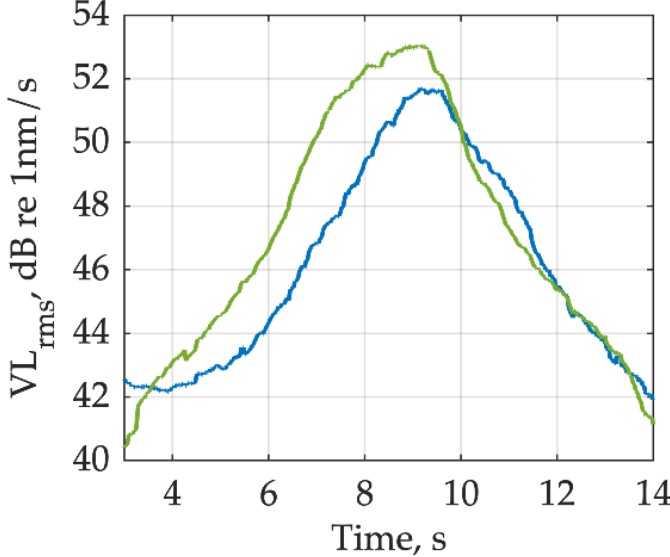

**Figure 6.** Particle velocity level $VL$ (blue line) and its estimate $\hat{V}L$ under the plane wave approximation (green line).

### 3.3. Directivity of the Hovercraft Underwater Noise

The following algorithm is applied to estimate the azimuthal dependence of sound pressure level produced by the hovercraft in a horizontal plane.

1. Simultaneous recording of the time-dependent sound pressure level $SPL(t)$ at a hydrophone, distance $r(t)$ between the hovercraft and a hydrophone, azimuthal angle $\theta(t)$ between the hovercraft heading, and the hovercraft–hydrophone direction.
2. Normalization of $SPL(t)$ to a nominal distance of 1 m using the spherical decay law, $SPL_1(t) = SPL(t) + 20\lg r(t)$.
3. Merging the time series of $SPL_1(t)$ and $\theta(t)$ to obtain angular dependence $SPL_1(\theta)$.

Since the angular dependence $SPL_1(\theta)$ is found, one can simulate a two-dimensional distribution of the hovercraft noise as

$$SPL(r, \theta) = SPL_1(\theta) - 20\lg r, \tag{9}$$

which can be converted to the Cartesian coordinates $(x, y)$.

Figure 7 shows an example of the spatial distribution $SPL$ at a given time during a hovercraft passage. The symmetry of radiation from the left and right sides of the hovercraft is assumed. Note that when the hovercraft is moving away from PGS, the monopole-type radiation prevails, while the forward radiation pattern is irregular that suggests the multipole-type radiation.

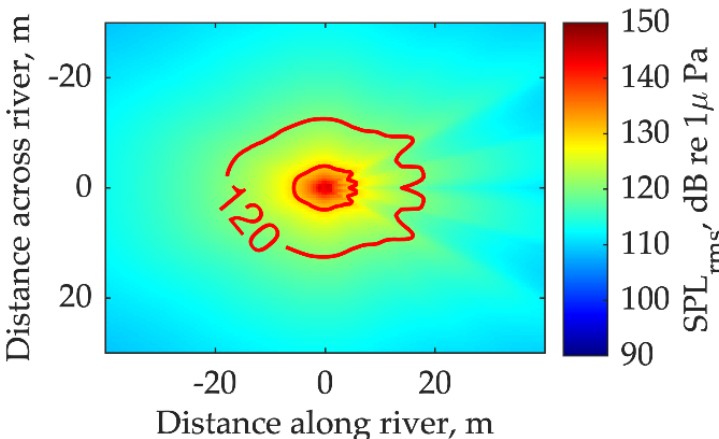

**Figure 7.** Directivity pattern of underwater noise produced by the hovercraft in the Ural-Caspian channel (heading is in the positive direction of the horizontal axis).

A zone of a potential negative hydroacoustic impact on the fish fauna is located inside the contour line of 130 dB re 1 μPa in terms of pressure (≈67 dB re 1 nm/s is the equivalent level for the particle velocity). According to Figure 6b, this zone is localized within a distance of up to ≈10 m from the center of the hovercraft. A negative impact of the underwater noise on the fish fauna may be expected only in the vicinity of the stern of the hovercraft.

*3.4. Comparison of the Underwater Noise Produced by the Hovercrafts at Different Speeds and by Other Vessels*

Table 1 provides the average sound pressure levels $SPL_{25}$, sound exposure levels $SEL_{25}$, and particle velocity levels $VL_{25}$ for various regimes of the hovercraft running. The levels are averaged over all field measurements conducted in the Ural-Caspian Channel, in the NCOSRB enclosed basin, and in the Northern Caspian Sea. All values are normalized to a distance of 25 m. Three speed regimes are considered (slow speed, cruising speed, and maximum speed). The minimum level of underwater noise was observed when hovercraft running at a cruising speed.

**Table 1.** Average hovercraft underwater noise levels (*SPL*, *SEL*, and *VL*) normalized to a distance of 25 m for various speeds. The RMS deviation for the presented values does not exceed ≈3 dB.

|  | $SPL_{25}$, dB re 1 μPa | $SEL_{25}$, dB re 1 μPa$^2$s | $VL_{25}$,dB re 1nm/s |
|---|---|---|---|
| Low speed (up to 7 m/s) | 115 | 126 | 50 |
| Cruising speed (7–15 m/s) | 110 | 116 | 45 |
| Maximum speed (more than 15 m/s) | 114 | 120 | 49 |

The field experiments included measurements of underwater noise produced by other vessels passing along the Ural-Caspian Channel. When towing a barge along the channel, the noise level of the "M3 141" towboat has a value of $SPL_{25} = 136.6$ dB re 1 μPa re 25 m. The towboat has a screw propeller. Underwater noise levels from various fishing and freight vessels with standard screw propellers can be fenound in [1–3]. In addition to the noise measurements from the hovercraft *Caspian Falcon*, the noise from a smaller hovercraft *Ermek* (class of ship Neptune 34 Irbis, length 17.6 m, $v_{max}$ 65 km/h) was analyzed. Another type of vessel operating in the North Caspian region is the shallow-draft water-jet vessels: the fast rescue craft (FRC) and the ultrashallow-draft freight vessel (SD1). Images and specifications

of these vessels are provided in the Supplementary Material [38]. The measured sound exposure levels $SEL_{25}$ for different speeds are shown in Figure 8. Note that the levels of the hovercrafts noise at any speed are significantly lower than those produced by the vessels with standard screw propellers or water-jet propulsion system.

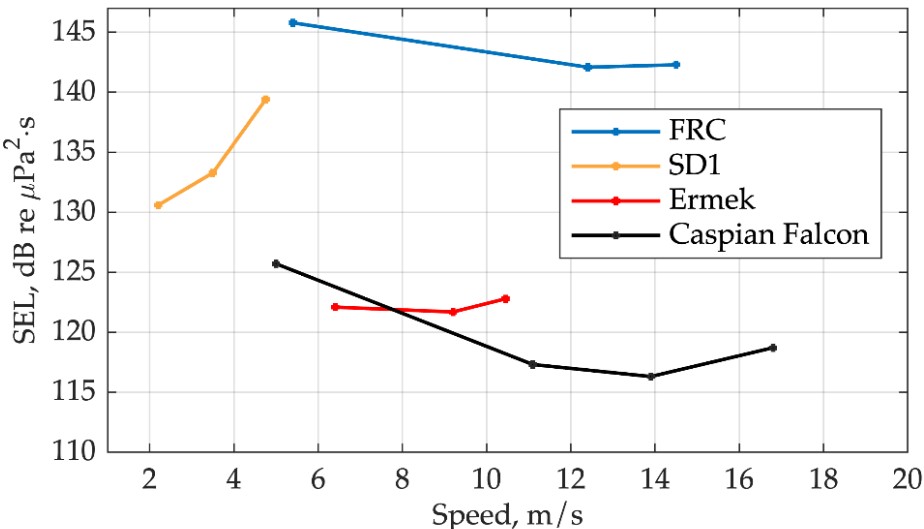

**Figure 8.** Sound exposure level $SEL_{25}$ of the hovercraft (black curve), SD1 vessel (red curve), and FRC vessel (blue curve). Values of $SEL$ are normalized to a CPA distance of 25 m.

*3.5. Hydrodynamic Pressure Surge under the Hovercraft Cushion*

Hydrodynamic pressure surge in water due to the running hovercraft was registered with a low-sensitivity Brüel & Kjaer 8104 hydrophone ($-205$ dB re. 1 V/µPa) connected to the RTsys recorder. The frequency band of this type of a hydrophone starts at 0.1 Hz. The hydrophone was deployed near the bottom at a depth of 4 m under the hovercraft track, i.e., the hovercraft was passing above the hydrophone.

Figure 9 shows the recorded pressure time series for seven passages of the hovercraft. A pressure surge (Figure 9c) when the hovercraft was passing directly above the hydrophone reaches 1 kPa (peak-to-peak) or 180 dB re 1 µPa, which is comparable in order of magnitude to the pressure created under the hovercraft skirt (flexible enclosure of cushion). After passing, noticeable pressure variations are observed for several minutes (Figure 9b), apparently, due to fluctuations in the wake layer. Note that the rate of rise and discharge of pressure is not high at a hydrophone placed near the bottom. The pressure variation is slow even at the maximum speed of the hovercraft. Moreover, the pressure surge level (180 dB re 1 µPa) and pressure rise rate are significantly lower than those of the pulses generated by an airgun (220–250 dB re 1 µPa at 1 m) or during pile driving. It means that the recorded pressure surges are not shock waves. Since the rise time of the pressure surge lasts a few seconds, a negative acoustic effect on the fish fauna is hardly possible, though hydrodynamic effects may be significant [39].

*3.6. Airborne Noise of the Hovercraft*

In the experiments, the background noise level did not exceed 50 dBA, and the average wind speed was less than 6 m/s. The airborne hovercraft noise levels exceeded the background values by more than 20 dBA.

Figure 10 shows the dependence of $SPL_{A25}$ on the hovercraft speed. The lowest values of $SPL_{A25}$ (90–95 dBA) are observed at a cruising speed of 7 to 14 m/s. The minimum value of $SEL_{A25}$ (96 dBA) also corresponds to this speed range (Figure 11). High noise levels at low speeds are likely associated with the episodes of a short-period propulsion engine RPM increase, which is necessary to compensate the wind drift of a hovercraft. Moreover, three points with the highest $SPL_{A25}$ (107–114 dBA) were obtained in the stern direction

of the hovercraft. It should be noted that the benefit of using cruising speed to minimize noise impact is confirmed by the measurement results from both years, since the noise at cruising speed (7–15 m/s) is the lowest (see Figures 10 and 11).

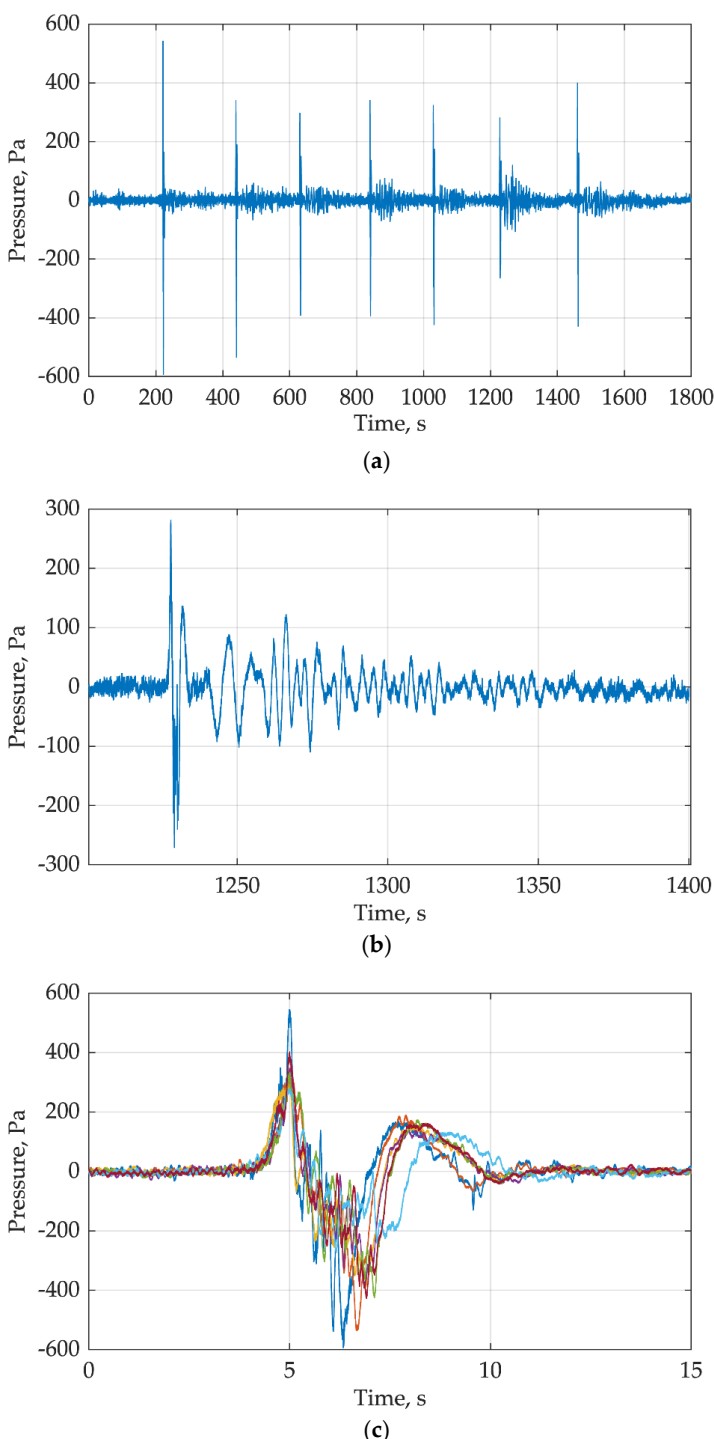

**Figure 9.** (**a**) Pressure time series measured under the hovercraft during its seven passages; (**b**) example of pressure time series for one passage (high amplitude hydrodynamic surge is observed with the further long-time fluctuations); (**c**) the scaled-up hydrodynamic surges from seven passages of the hovercraft.

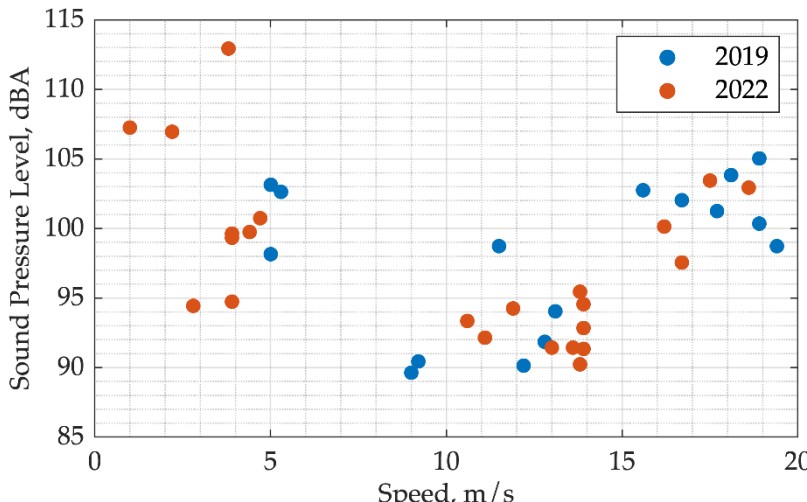

**Figure 10.** Dependence of the airborne noise pressure level, $SPL_{A25}$, in dBA on the hovercraft speed. Red markers show the data of 2022; blue markers show the data of 2019.

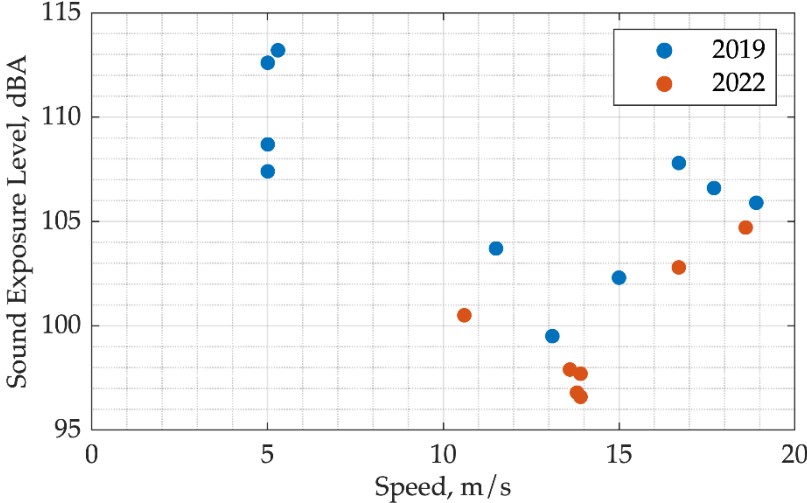

**Figure 11.** Dependence of the airborne noise exposure level, $SEL_{A25}$, in dBA on the hovercraft speed. Red markers show the data of 2022; blue markers show the data of 2019.

To estimate the safety zone boundary for the bird fauna in the estuary of the Ural River, two-dimensional distributions of the hovercraft airborne noise in a horizontal plane are calculated according to the ISO Recommendations [40] using the following formula:

$$SPL_A(r, \theta) = SPL_{A1}(\theta) - 20\lg r - A_{gr} - A_{atm} + 3, \qquad (10)$$

where $A_{atm} = r/1000$, $A_{gr} = \begin{cases} 0, r < 100m \\ 4.8 - \frac{10}{r}\left(17 + \frac{300}{r}\right), r > 100m \end{cases}$ are the adjustments made to account for the sound attenuation in the air and reflection from the water surface; $SPL_{A1}(\theta)$ is an angular dependence of the noise level normalized to a distance of 1 m. To calculate $SPL_{A1}(\theta)$, the same procedure as used for the underwater noise is implemented.

Figure 12 shows an example of the airborne noise level $SPL_A(r, \theta)$ in a horizontal plane during running of the hovercraft in the Ural-Caspian Channel at a speed of 13.6 m/s. Radiation pattern of the hovercraft noise in the air has a few features: (1) relatively uniform radiation from the sides with some attenuation due to the presence of propeller cowlings, (2) an increased level towards the stern, and a decreased level towards the nose due to the partial shading of the noise by the deckhouse.

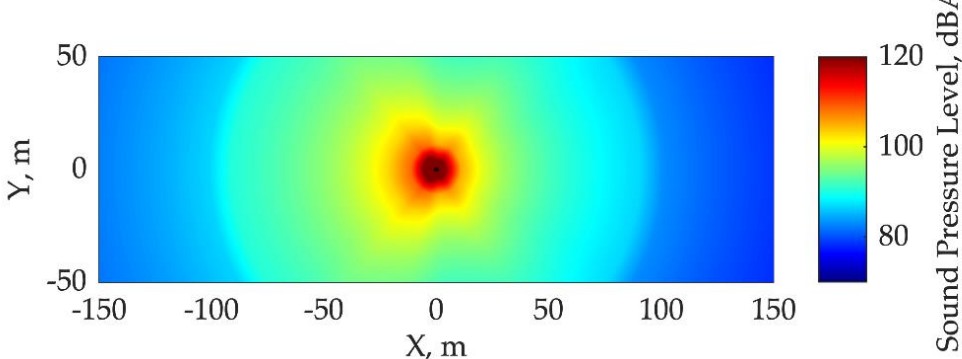

**Figure 12.** Hovercraft airborne noise level $SPL_A(r,\theta)$ in a horizontal plane at a cruising speed of 13.6 m/s. The hovercraft is passing rightwards.

Contours of the noise level $SPL_A(r,\theta)$ from Figure 12 are also plotted on a terrain map in Figure 13. Two assumptions are made when plotting Figures 12 and 13. The first one is the left–right symmetry of noise radiation by a hovercraft. The second one is that the terrain is flat. Figure 13 (an example of the "noise map") shows that levels of the hovercraft airborne noise do not exceed the threshold values of 80 dBA starting from just beyond the shoreline. The level of 80 dBA is an onset of the negative response of birds according to the literature data [41]. Such kind of "noise maps" plotted for different hovercraft speeds can be useful for ornithologists studying the onset of a negative bird's reaction to the hovercraft noise. To do this, it is enough to have the "noise maps" for different speeds of hovercraft and GPS coordinates of the bird's site and of the hovercraft. In this case, the distance to the hovercraft is estimated from GPS data when negative reactions of birds are observed. Then, the level of airborne noise is determined from the "noise maps" simulated for different distances and speeds the known position and speed of the hovercraft. It can help in catching up on the onset moment of a negative bird's reaction to the level of hovercraft noise without noise measurements in situ.

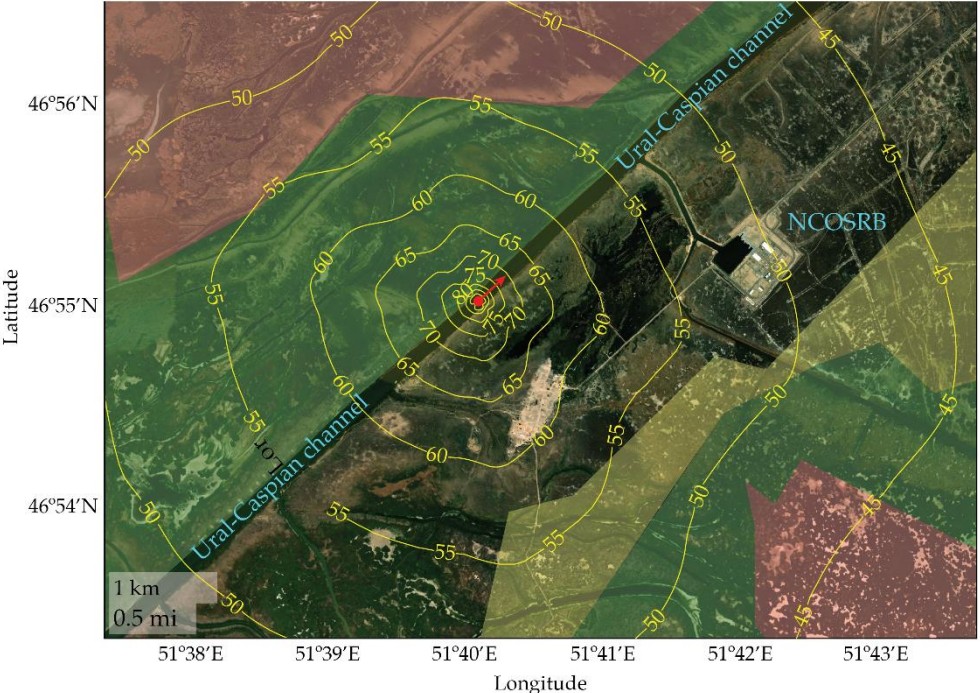

**Figure 13.** Example of the "Noise map"— contours of the noise level $SPL_A(r,\theta)$ from Figure 11 in the Ural-Caspian Channel. Red arrow denotes the hovercraft movement direction.

## 4. Discussion and Conclusions

Experimental underwater and airborne noise levels of Griffon BHT130 hovercraft *Caspian Falcon* presented in Table 2 demonstrate the maximum values for the hovercraft running at minimum and maximum speeds and the lowest values for cruising speed. For any speed, the underwater noise is substantially lower than that radiated by the vessels with screw or water-jet propellers. The airborne noise levels are higher than those of propeller vessels. However, when the hovercraft is moving at the cruising speed (7–15 m/s) in the middle of the Ural-Caspian Channel, the noise level just beyond the shoreline does not exceed the threshold value of 80 dBA for the negative response of birds [41].

**Table 2.** Averaged sound pressure levels of the hovercraft noise underwater and in the air normalized to a distance of 25 m.

| | $SPL_{25}$ of Underwater Noise (dB re. 1 μPa) at a Distance of 25 m | $SPL_{A25}$ of Airborne Noise (dBA re. 20 μPa) at a Distance of 25 m |
|---|---|---|
| Low speed (up to 7 m/s) | 115 | 103 |
| Cruising speed (7–15 m/s) | 110 | 93 |
| Maximum speed (more than 15 m/s) | 114 | 102 |
| Background | 99 | 50 |

Field measurements of underwater and in-air sound in the Ural-Caspian Channel and in the North Caspian Sea revealed the following features of the noise emission from the hovercraft *Caspian Falcon*.

1.  There is a strong dependence of the sound pressure level, sound exposure level, and particle velocity level on the hovercraft speed, with minimums at the cruising speed. Using the plane wave approximation, the particle velocity level of underwater noise can be retrieved from the sound pressure level with an accuracy of ≈3 dB at the range of ≈ 20–50 m in the frequency band 100–1650 Hz.
2.  Underwater noise level ($SPL_{25}$ or $SEL_{25}$) variation does not exceed ≈3 dB depending on the site of measurements and environmental conditions. This indicates a similarity of waveguide parameters at various points in the Ural River Estuary.
3.  Nonuniform noise radiation pattern in a horizontal plane is observed both underwater and in the air. This pattern is necessary to know for the accurate prediction of safety zones for the fauna.
4.  Hydrodynamic pressure surge under the hovercraft cushion at the maximum speed is about 1 kPa at 4 m depth. Its effect on the fish fauna is a matter of further research.

The simulated airborne "noise maps" (similar to those shown in Figure 13) coupled with the hovercraft GPS coordinates are very useful for studying the threshold of a negative response of birds to the airborne hovercraft noise even if parallel in situ noise measurements are not available. This is the original insight on how to investigate and control the hovercraft noise impact on the wildlife in the North Caspian Sector.

The results of the present study show that using a large hovercraft can be safe for ornitho- and fish fauna in the estuary of the Ural River and the Kazakhstan sector of the North Caspian which are a unique wildlife preservation. A cruising speed interval of 7–15 m/s is recommended as an optimum to minimize an acoustic impact from hovercrafts on ornitho- and fish fauna.

Future research should include the investigation of the correlation between the underwater and airborne components of hovercraft noise as well as a numerical and theoretical study of the hydrodynamic pressure surge effect under the hovercraft cushion.

**Supplementary Materials:** The following supporting information can be downloaded at https://www.mdpi.com/article/10.3390/jmse11051079/s1.

**Author Contributions:** Project Management: A.I.V.; Work Program, Methodology: A.I.V. and S.S.K.; Preparation of equipment; data collection, analysis and interpretation: O.Y.K., A.A.L. and A.S.S.; writing and editing of paper: A.I.V., O.Y.K., A.A.L. and S.S.K.; translation of paper into English: S.S.K.; funding acquisition: S.S.K. All authors have read and agreed to the published version of the manuscript.

**Funding:** This research was funded by North Caspian Operating Company N.V. (NCOC N.V) grant number JO 229077.

**Institutional Review Board Statement:** The experiments during which this data was collected were conducted according to the guidelines of the Declaration of Helsinki.

**Informed Consent Statement:** Not applicable.

**Data Availability Statement:** The acoustic data supporting this study is available on request from the North Caspian Operating Company N.V.

**Acknowledgments:** The authors are grateful to the Personnel of RAS Institute of Oceanology; PGS Designer, D.A. Shvoyev, and acoustician, A.V. Shatravin, member of the experiments held in 2017 and 2019; the crew of SD and FRC vessels of Veritas Marine Company; crew members of *Caspian Falcon* vessel of Caspian Offshore Construction Company; personnel of NCOSRB of KMG Systems & Services LLP who ensured implementation of the experiments.

**Conflicts of Interest:** The authors declare no conflict of interest. The funders had no role in the collection, analyses, or interpretation of data; in the writing of the manuscript, or in the decision to publish the results.

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
