# Peer review of "Airborne and Underwater Noise Produced by a Hovercraft in the North Caspian Region: Pressure and Particle Motion Measurements"

_jmse, doi:10.3390/jmse11051079_

Round 1
Reviewer 1 Report
(1)I suggest revising the abstract of the paper. It is necessary to highlight the main work, innovative points, and scientific significance of the paper in the abstract.
(2)It is recommended to explain the key points of this paper in the “Introduction”, indicating the differences between hovercraft noise and conventional ship noise, helicopter noise, etc. Highlight the characteristics of this article.
(3)Suggest adding testing methods and experimental conditions in “1Field procedures and acoustic equipment ”, as well as how to remove the influence of environmental factors.
(4)The data results are presented in “2. Underwater Noise”. It is recommended to increase in-depth analysis of the data results and summarize the laws behind the measurement results.There is also the same in “3. Hydrodynamic Pressure Surge under the Hovercraft Cushion”.
(5)I suggest modifying and improving the conclusion. Highlight the results and their innovation in the conclusion, and indicate their significance for scientific research and engineering applications.
(1)The layout of the figures in the article is messy, and it is recommended to make modifications and improvements.
(2)Further modifications can be made to improve the presentation and wording of the paper.
Reviewer 2 Report
Review Report on “Airborne and underwater noise produced by a Hovercraft IN the NORTH CASPIAN REGION: pressure and particle motion measurements” by Alexandr Vedenev et al.
The study focuses on measuring airborne and underwater noise radiated by a Griffon BHT130 hovercraft, which is being used for cargo and crew transportation in the environmentally sensitive area of the Ural River estuary. The article presents the results of several field campaigns organized from 2017 to 2022 to measure and analyze acoustic noise levels at various sites and hovercraft speeds. The directivity of the hovercraft noise was estimated and utilized for noise mapping of the Ural-Caspian Channel.
The article provides a comprehensive background of the research and outlines the need for the study. The authors clearly state their objectives and present a well-organized literature review.
The study provides valuable insights into the acoustic characteristics of hovercraft noise in the Ural-Caspian Channel and the North Caspian Sea. The authors have used an appropriate methodology to measure airborne and underwater noise levels, and the results are presented clearly. The discussion and conclusions are concise and summarize the main findings of the study adequately.
Overall, the article is an interesting study and is of current interest in the field of Marine Ecology. This is a worthwhile piece of research in terms of technical aspects and addresses some important aspects in the field. The simulations seem to be valid and significant to understand the research. However, I would suggest revising the article thoroughly thereby improving the presentation considerably and addressing the following points:
1. The introduction lacks a clear research question that could have guided the research and provided a better framework for the discussion.
2. The discussion lacks a critical analysis of the findings and their implications for the environment and the wildlife in the region. The authors do not provide any recommendations to mitigate the negative impact of hovercraft noise on the fauna habitat in the area.
3. The study could have benefited from a comparison with other noise sources in the region and a more detailed analysis of the hydrodynamic effects of the passing hovercraft.
4. Additionally, the authors have not discussed the limitations of the study, such as the small sample size and the restricted area of the measurements.
5. The conclusion is concise and summarizes the main findings of the study, but it does not offer any new insights or future directions for research.
6. Proofread the whole article carefully for grammar and spelling errors.
Overall, the study provides a useful contribution to the understanding of the acoustic characteristics of hovercraft noise in the Ural-Caspian region. However, the authors could have provided a more comprehensive and critical analysis of the results and their implications, thereby addressing the above stated points.
Best wishes
Largely, the English language appeared to be fine. However, it requires minor editing in addressing the grammatical and typos errors.
Reviewer 3 Report
The reported study is a real interesting, and one of a few only, paper investigating simultaneously both underwater and airborne ship noise emissions. Whilst the investigated source is only one, the paper surely deserves attention as it is follows a double approach that should be followed also by other authors, as well as the authors are encouraged to follow their studies on different ships. Details are reported below.
Formatting of the text should comply with journal’s rules. Please check it carefully as there are many errors.
Introduction is poor, as well as references. They should be better focused on both the investigated aspects, trying to balance the weight of them. About airborne noise, which seems to be the less mentioned, I suggest to add a period like:
“in the recent years investigations have been finally dedicated to airborne sound emitted by moving ships and port activities (Bernardini, M.; et al. Noise Assessment of Small Vessels for Action Planning in Canal Cities. Environments 2019, 6, 31.; Fredianelli, L.; et al., Pass-by characterization of noise emitted by different categories of seagoing ships in ports. Sustainability. 2020, 12(5), 1740; Nastasi, Marco, et al. "Parameters affecting noise emitted by ships moving in port areas." Sustainability 12.20 (2020): 8742; Badino, Aglaia, et al. "Airborne noise emissions from ships: Experimental characterization of the source and propagation over land." Applied Acoustics 104 (2016): 158-171; Borelli, Davide. "Maritime Airborne Noise: Ships and Harbours." International Journal of Acoustics and Vibration, vol. 24, no. 4, Dec. 2019, p. 631; Schiavoni, Samuele, et al. "Airborne Sound Power Levels and Spectra of Noise Sources in Port Areas." International Journal of Environmental Research and Public Health 19.17 (2022): 10996; Fredianelli, Luca, et al. "Source characterization guidelines for noise mapping of port areas." Heliyon 8.3 (2022): e09021; Fredianelli, Luca, et al. "Classification of noise sources for port area noise mapping." Environments 8.2 (2021): 12.) recognized as disturbing sources on citizens (Licitra, Gaetano, et al. "Port noise impact and citizens’ complaints evaluation in RUMBLE and MON ACUMEN INTERREG projects." Proceedings of the 26th International Congress on Sound and Vibration, Montreal, QC, Canada. 2019.).
Use dB(A).
Please elaborate more about sound emissions and speed dependance, but also compute the sound power level as a possible input to noise models. Suggestions on the procedure can be taken from Fredianelli, L.; et al., Pass-by characterization of noise emitted by different categories of seagoing ships in ports. Sustainability. 2020, 12(5), 1740; Nastasi, Marco, et al. "Parameters affecting noise emitted by ships moving in port areas." Sustainability 12.20 (2020): 8742.
Conclusions should better summarize the results.
My suggestion, at least for future work, is to investigate for correlation between the underwater and airborne components. This can be mentioned in a future developments part of the conclusions.
Avoid use of “we” in scientific writing.
English is fine, but avoid the use of "we" in scientific formal writing
Round 2
Reviewer 1 Report
(1)The meaning of each line is not given in Figure 4 (a). In addition, the trends of these lines are very similar. It is recommended to use a better form of data expression to highlight the meaning of that figure. It is recommended to pay attention to similar issue of figures in the paperm, such as figure4(c), figure 5(a), and so on.
(2)The results of table1 indicate that the higher or lower the velocity, the greater the sound pressure value, which is somewhat different from conventional understanding. Can you further explain the rationality of the results?
(3)I suggest adding the further analysis and explanation of Figure 9, including the relationships between the various charts, the characteristics of the data in the charts, etc.
(4)The data in Figures 10 and 11 indicate significant differences between 2019 and 2022. It is recommended to add the reasons for these differences and demonstrate the rationality of the differences.
(5)The research in this article focuses on practical measurements, and it is recommended to include analysis of existing work shortcomings, feasibility of numerical simulation, and further work that can be carried out in the future in the Discussion. This can point out the direction for further in-depth research in the future.
(1)Text and charts can be further improved.
Reviewer 3 Report
The authors improved a bit the manuscript, but there are still some aspects to be fixed:
Please continue to revise a bit introduction, as previously suggested. The paragraph can still be improved.
Most important, chapter discussions and conclusions must be fixed: no equation should be reported in the conclusions, as well as plain text is better than bullets. Thus, my suggestion is to divide discussions and conlcusions, and enrich the 2 separate parts. Discussion should compare results with other studies (previously suggested), and conclusions should summarize the overall work, with a focus also on future developments and limitations.
can still be improved
Round 3
Reviewer 1 Report
No
No